# Possible Reaction Mechanisms Involved in Degradation of Patulin by Heat-Assisted Cysteine under Highly Acidic Conditions

**DOI:** 10.3390/toxins14100695

**Published:** 2022-10-10

**Authors:** Enjie Diao, Kun Ma, Minghua Li, Hui Zhang, Peng Xie, Shiquan Qian, Huwei Song, Ruifeng Mao, Liming Zhang

**Affiliations:** 1Jiangsu Collaborative Innovation Center of Regional Modern Agriculture & Environmental Protection, Huaiyin Normal University, Huai’an 223300, China; 2Jiangsu Key Laboratory for Food Safety & Nutrition Function Evaluation, Huaiyin Normal University, Huai’an 223300, China; 3College of Food Science & Engineering, Shandong Agricultural University, Tai’an 271018, China; 4School of Pharmacy, Jiangsu Food and Pharmaceutical Science College, Huai’an 223003, China; 5Research & Development Center of National Vegetable Processing Technology, Jiangsu Liming Food Group Co., Ltd., Pizhou 221354, China

**Keywords:** patulin, cysteine, degradation product, Michael addition reaction, toxic effect

## Abstract

Patulin (PAT) is one of mycotoxins that usually contaminates apple juice, and it is not easily detoxified by cysteine (CYS) at room temperature due to the highly acidic conditions based on the Michael addition reaction. However, it could be effectively degraded by a heating treatment at 120 °C for 30 min in the presence of cysteine. In our study, a total of eight degradation products (DP A–H) were characterized and identified via liquid chromatography quadrupole time-of-flight mass spectrometry (LC-Q-TOF-MS) in a negative ion mode, and their structures and formulas were proposed based on their accurate mass data. The fragmentation patterns of PAT and its degradation products were obtained from the MS/MS analysis. Meanwhile, the possible reaction mechanisms involved in the degradation of PAT were established and explained for the first time. According to the relation between the structure and toxicity of PAT, it could be deduced that the toxic effects of PAT degradation products were potentially much less than those of PAT-self.

## 1. Introduction

Patulin (PAT) is one of the mycotoxins that usually contaminates apple juice and is a polyketide secondary metabolite mainly produced by *Penicillium expanisum* [1,2,3,4]. Results from animal and cell culture experiments in toxicology have verified the carcinogenicity, teratogenicity, mutagenicity, genotoxicity, and immune toxicity of PAT [5,6]. In order to protect the health of consumers, WHO and EU have set the maximum levels of PAT at 50 μg/L for apple juice, 25 ng/g for solid apple products, and 10 ng/g for infants and young children’s foods [7,8].

According to our investigation, PAT in apple juice is removed mainly by food processing (milling, pasteurization, enzymatic treatment, microfiltration, evaporation, etc.) and physical adsorption (clarification) using activated carbon or macroporous resins in the juice industry [9]. Presently, other physical, chemical, or biological methods for detoxifying PAT have not been adopted for commercial application in the juice industry due to some drawbacks, such as the lower detoxification efficiency, higher cost in detoxifying PAT, larger damage to the nutrition, and sensory quality of juice, easily causing cross-contamination between the juice and the environment by undestroyed toxin, and unrealized industrial application due to the limited conditions [9].

However, food processing and physical adsorption (clarification) for removing PAT also present some disadvantages. Food processing could not completely remove or degrade PAT in juice, and has even caused cross-contamination due to the improper operation during processing [10]. Physical adsorption (clarification) significantly reduced the phenolic compounds and organic acids in apple juice; meanwhile, the high level of PAT remained in pulps, which could be harmful to animals as feed [11]. In addition, PAT was not destroyed during the physical adsorption (clarification) process, which remained in adsorbents or precipitants, and then could cause environment pollution if they were directly discarded without detoxification treatment.

In identifying an ideal method for removing or detoxifying PAT in juice, we found that cysteine may be a very good detoxification agent [12]. Cysteine contains a thiol-group (–SH), which could spontaneously react with PAT to form the covalent adducts (PAT–CYS) at near-neutral pH (pH 7.4) due to the reactivity of thiolate ions [13]. The thiol group of cysteine was attached to one of the double bonds of PAT based on the Michael addition reaction (a nucleophilic addition reaction) [14]. It is well-known that the pH values of apple juice generally range from 3.0 to 4.0, which is a very difficult range for degrading PAT by cysteine based on the nucleophilic addition reaction at room temperature under the strong acidic conditions (pH < 4.0) [15]. Therefore, we used the heat-assisted cysteine to degrade PAT under the highly acidic conditions (pH 3.5), and found that this method could effectively degrade it [16,17]. However, the degradation products (DPs) of PAT by the heat-assisted cysteine have not been identified, and the degradation reaction mechanisms of PAT by heat-assisted cysteine under the highly acidic conditions have not been investigated, which limits the commercial application of this detoxification technology. The main objective of this study is to identify the DPs of PAT by liquid chromatography quadrupole time-of-flight mass spectrometry/mass spectrometry (LC-Q-TOF-MS/MS) analysis and deduce the possible reaction mechanisms involved in the degradation of PAT by heat-assisted cysteine under the highly acidic conditions (pH 3.5).

## 2. Results

### 2.1. Degradation of PAT by Heat-Assisted Cysteine under Highly Acidic Conditions

The degradation of PAT by heat-assisted cysteine under highly acidic conditions is shown in Figure 1 and Appendix A (see the Appendix A). PAT in pH 3.5 of the simulated juice solution was only reduced by 19.19~38.43% after heating treatment at 120 °C for the times ranging from 30 to 90 min in the absence of cysteine. In the presence of cysteine, PAT was almost completely degraded, and the degradation rate was greater than 99.9% after heating treatment at 120 °C for 30 min.

### 2.2. Fragment Patterns of PAT Based on the LC-Q-TOF-MS Analysis

PAT and its DPs obtained from the treatment of heat-assisted cysteine under highly acidic conditions were subjected to the LC-Q-TOF-MS analysis. Their structural characterizations were studied based on the *m/z* ratio and mass fragmentation patterns, and their structures were identified according to the accurate mass measurement and calculating ppm error.

PAT consists of two main rings, i.e., the pyran ring and furo-lactone ring. The MS fragment pattern of PAT was proposed based on the LC-Q-TOF-MS analysis, and its MS/MS spectra and data are shown in Figure 2 and Table 1, respectively. The molecular ion peak is labeled as [M–H]^–^. PAT showed a molecular ion at *m/z* 153.0197 (C_7_H_5_O_5_^–^) in the negative ion mode. The main fragment ions formed were at *m/z* 125.0215 (losing the carbonyl group on the lactone ring), *m/z* 111.0078 (losing the acetyl group on the lactone ring), and *m/z* 109.0260 (losing the carboxyl group on the lactone ring). The molecular ion at *m/z* 111.0078 has two structures with tautomer (i.e., keto–enol tautomerism). The fragment patterns of PAT are displayed in Figure 2. The elemental composition was confirmed by the accurate mass measurement, as mentioned in Table 1. Table 1 shows the most probable molecule formulas, retention times, observed and calculated mass values, errors, numbers of the double bond equivalent (DBE, i.e., unsaturation degrees), scores, and major fragment ions for PAT and its DPs.

### 2.3. Identification of PAT Degradation Products Based on the LC-Q-TOF-MS Analysis

A total of eight DPs (DP A–H) were identified and characterized based on the accurate mass, error ppm, and DBE. The MS/MS spectrum of PAT DPs (DP A→H) are shown in Figure 3, which is showing the major fragment ion peaks. According to the changes in *m/z* ratios of product ions and their precursors, as well as those of PAT and cysteine, the structures and formulas of PAT DPs were proposed, which are listed in Table 1. Table 1 also shows the most probable molecule formulas, retention times, observed and calculated mass values, errors, the DBE, scores, and major fragment ions for PAT DPs (DP A–H). The structures of PAT and its DPs are shown in Figure 4.

Figure 5 presents the liquid chromatography coupled with diode array detector (LC-DAD) chromatograms of PAT and its DPs in the simulated juice solution. As can be seen from Figure 5, only six DPs of PAT were detected, namely, DPs A, C, D, E, F, and H. The reason may be that the DPs B and G of PAT have no UV absorption or very weak absorption at 276 nm, which made them unable to be detected. The peaks appearing before 1.2 min are the ones of cysteine and some impurities.

#### 2.3.1. Degradation Product A (DP A)

The parent ion peak [M–H]^–^ of DP A was at *m/z* 256.0823 with a plausible chemical formula C_10_H_11_NO_5_S. Figure 3A shows the concise fragment pattern for the DPs of PAT, which reveals the major product ions of DP A at *m/z* 136.0399, *m/z* 150.0538, and *m/z* 162.0557 (Table 1). These product ions were formed by eliminating C_3_H_4_O_3_S, C_2_H_2_O_3_S, and C_5_H_2_O_2_ from the parent ion C_10_H_11_NO_5_S (DP A), respectively (Appendix A).

#### 2.3.2. Degradation Product B (DP B)

The DP B has a parent ion peak [M–H]^–^ at *m/z* 212.0388 (C_9_H_11_NO_3_S). Its characteristic fragment peaks were seen at *m/z* 172.0422, *m/z* 154.0305, *m/z* 138.0560, *m/z* 126.0560, and *m/z* 112.0402 (Figure 3B, Table 1). They were formed by losing C_4_H_4_, C_2_H_2_S, C_2_H_2_OS, C_4_H_6_S, and C_4_H_4_O_3_ from the parent ion of DP B, respectively (Appendix A).

#### 2.3.3. Degradation Product C (DP C)

The DP C presented an ion fragment at *m/z* 377.0475 [M–H]^–^ (Figure 3C). It dissociated into the fragment ions of *m/z* 290.0826, *m/z* 273.0051, *m/z* 182.0254, and *m/z* 168.0313 on the elimination of C_3_H_5_NO_2_, C_3_H_4_O_2_S, C_6_H_6_NO_3_S, and C_7_H_8_NO_3_S from the parent ion of DP C, respectively (Table 1, Appendix A).

#### 2.3.4. Degradation Product D (DP D)

The DP D showed its parent ion peak [M–H]^–^ at *m/z* 349.0530 (Figure 3D). The prominent fragment peaks from DP D were seen at *m/z* 305.1421, *m/z* 150.0556, and *m/z* 120.0121 by the successive losses of CO_2_, C_5_H_13_NO_5_S, and C_9_H_11_NO_4_S (Table 1, Appendix A). The fragment ion at *m/z* 305.1421 has two isomers, which formed by losing CO_2_ (decarboxylation) from any one of the two cysteine molecules bound on DP D (Appendix A).

#### 2.3.5. Degradation Product E (DP E)

The parent ion peak [M–H]^–^ of DP E was observed at *m/z* 246.0439 (Figure 3E). Three main fragment ions were at *m/z* 150.0556, *m/z* 138.0557, and *m/z* 123.0318 (Table 1). Both the fragment ions at *m/z* 138.0557 and *m/z* 123.0318 each have two isomers, and both of them were keto-enol tautomerism (Appendix A).

#### 2.3.6. Degradation Product F (DP F)

The accurate mass of DP F ion was *m/z* 331.0424 with elemental composition C_12_H_16_N_2_O_5_S_2_ (Figure 3F), which was dissociated to give two main fragment ions of *m/z* 200.0204 and *m/z* 166.0328 with the loss of C_4_H_5_NO_4_ and C_4_H_7_NO_4_S from the parent ion of DP F, respectively (Table 1, Appendix A).

#### 2.3.7. Degradation Product G (DP G)

A parent ion peak [M–H]^–^ at *m/z* 313.0319 represented the DP G (Figure 3G), which also yielded two main fragment ions of *m/z* 282.0426 and *m/z* 193.0804 with the loss of CH_5_N and C_3_H_4_O_3_S, respectively (Table 1, Appendix A).

#### 2.3.8. Degradation Product H (DP H)

The parent ion peak [M–H]^–^ of DP H was detected at *m/z* 217.0441 (Figure 3H). Its characteristic fragment ions were observed at *m/z* 204.0091, *m/z* 174.0264, *m/z* 131.0359, and *m/z* 117.0453 by the successive losses of CH, CO_2_, C_3_H_4_NO_2_, and C_5_H_8_O_2_ (Table 1, Appendix A). The fragment ion at *m/z* 204.0091 has three isomers; two of these were keto-enol tautomerism (Appendix A) and the third was position isomer (different position of the double bond).

### 2.4. Degradation Mechanism of PAT by Heat-Assisted Cysteine under Highly Acidic Conditions

The most possible degradation mechanism of PAT by heat-assisted cysteine under the highly acidic condition is outlined in Figure 6. Under the high temperature (120 °C) and acidic conditions (pH 3.5), firstly, 1 mole of PAT (PAT) reacted with 1 mole of cysteine (CYS) to form the PAT–CYS adduct based on the Michael addition reaction, which belongs to nucleophilic addition reactions [14]. The CYS molecule was added to the C2–C3 double bond of PAT. The formed PAT–CYS adducts continued to be degraded or reacted with CYS under the high temperature and acidic environment via two branches. The first branch was that PAT-CYS adducts lost H_2_O by the esterification reaction between the carboxyl group (–COOH) on the CYS and the hydroxyl group (–OH) of C4 on the PAT, and yielded the DP A. The single bond on the C7–O of DP A was attacked by H^+^ under the high temperature and acidic conditions, and resulted in the opening of the lactone ring, which quickly lost CO_2_ by the decarboxylation reaction and formed the DP B. The second branch was that the PAT–CYS adduct continued to bind 1 mole of CYS on the C6–C7 double bond to yield the PAT–2CYS adducts. A part of PAT–2CYS adducts underwent the intramolecular esterification to form the DP C with the loss of H_2_O, as in the formation of DP A. Another part of the PAT–2CYS adduct was degraded to yield the DP D by opening the lactone ring and removing the CO_2_ based on the decarboxylation reaction, as in the formation of DP B. It is more complicated for the further degradation of DP D with at least three degradation pathways. The first is the formation of DP E obtained from the hydrolysis of DP D with the loss of CYS, and the DP E has two isomers, i.e., keto–enol tautomerism. The second is that DP D was successively converted into DP F and DP G by the esterification and the condensation reactions with loss of 2H_2_O under the high temperature and acidic conditions. The third is that DP D changed to an intermediate (C_12_H_22_N_2_O_6_S) firstly by adding the H at the double bonds on C2–C3 and C6–C7 based on the reduction reaction under high acidic conditions, and then the intermediate was rapidly dissociated into the DP H with the help of a high temperature at 120 °C.

Seen from the Figure 6, the degradation mechanisms of PAT by heat-assisted CYS under the highly acidic condition involved the Michael addition reaction, decarboxylation reaction, hydrolysis reaction, esterification reaction, condensation reaction, reduction reaction, thermal dissociation reaction, et al. The Michael addition reaction is the primary reaction process of PAT degradation.

## 3. Discussion

### 3.1. Degradation Efficiency of PAT with and without Cysteine

In this study, the LC-DAD method was used to determine the degradation efficiency of PAT with and without cysteine, which is the most-used chromatographic technique for PAT analysis due to its easy identification and quantification of PAT through its characteristic absorption spectrum [16,17]. 

Theoretically, one mole of PAT can react with two moles of cysteine based on the Michael addition reaction at room temperature under near neutral conditions [13]. In our reported literature, PAT could not be effectively decomposed by cysteine in highly acidic conditions (pH 3.5) when the reaction temperature was less than 60 °C [18]. Meanwhile, PAT is stable to heat in an acidic environment, and is very difficult to be degraded in the absence of cysteine [18,19]. To effectively degrade PAT, excess cysteine was used in this study, i.e., a molar ratio of 1:3 between PAT and cysteine was used. As can be seen from the Figure 1, PAT could be completely degraded by the heat-assisted cysteine at 120 °C under highly acidic conditions, while it was only reduced by 19.19~38.43% in the absence of cysteine under the other same conditions. The results further verified the synergistic role of both cysteine and heating (high temperature) on the degradation of PAT in highly acidic condition. Therefore, as a combined physical and chemical strategy, heat-assisted cysteine technique may be a good idea for use in the detoxification of patulin in the juice industry due to its advantages regarding detoxification efficiency, cost, operation, cross-contamination and environment pollution, and labor intensity.

### 3.2. Identification of PAT Degradation Products

LC-Q-TOF-MS/MS has been successfully used to analyze and identify metabolites and degradation products of food contaminants or organic substances by other groups based on its accurate mass measurement, error, and DBE [20,21,22,23]. In this study, eight of the PAT degradation products (DPs A–H) were characterized and identified by the LC-Q-TOF-MS/MS analysis, which were named based on where they appeared in the possible degradation pathway of PAT (Figure 6). While some DPs of PAT were not separated sufficiently by liquid chromatography (Figure 5), this could not affect the identification of these DPs, as the LC-Q-TOF-MS/MS has the high sensitivity, large mass-range, and simultaneous detection of ions of all masses in identifying organic compounds [20,21,22,23]. Moreover, the parent ions and fragment ions of PAT DPs have fixed retention times and an accurate *m/z* ratio, so they can be clearly identified based on their chromatographic peaks and accurate mass data (Figure 5, Table 1).

As can b seen from Figure 4, these products were PAT–CYS or PAT–2CYS adducts based on the Michael addition reaction, or some products obtained from PAT–CYS or PAT–2CYS adducts by the decarboxylation reaction, hydrolysis reaction, esterification reaction, condensation reaction, reduction reaction, thermal dissociation reaction, etc. (Figure 6). These product structures were closely related to the ones of patulin and cysteine (Figure 4).

### 3.3. Degradation Mechanisms of PAT

According to the reported literature, the Michael addition reaction can be carried out spontaneously at the room temperature and near-neutral conditions (pH 6.4~7.4) due to the electrophilic properties of PAT [14]. It is well-known that PAT is very stable to heat, especially in acidic pH [18,19,24]. In the presence of CYS, PAT in pH 3.5 solution was only reduced by 6.22% and 7.69% after being treated at 30 °C and 60 °C for 120 min, respectively [18]. Its degradation efficiencies were increased to 42.12, 78.94, and 100.00% after being heated at 90 °C, 120 °C, and 150 °C for 90 min, respectively [19]. It indicates that a temperature higher than 90 °C could promote the PAT degradation in the presence of CYS under the highly acidic conditions. The main reason may be that the high temperature (>90 °C) accelerated the moving and mutual collision vigorously between PAT and CYS molecules based on the collision theory of chemical reaction, and then promoted the formation of PAT–CYS and PAT–2CYS adducts based on the Michael addition reaction. Meanwhile, the high temperature also sped up the opening of the lactone ring on the PAT structure, and then promoted the decarboxylation reaction, hydrolysis reaction, condensation reaction, thermal dissociation reaction, et al., which further increased the formation of PAT–CYS and PAT–2CYS adducts. Therefore, the Michael addition reaction is the critical mechanism in the initial step of PAT degradation, and the following serial reactions facilitate the continuation of the Michael addition reaction.

### 3.4. Toxicity Prediction of PAT and Its Degradation Products

Many scholars have investigated the toxic effects of PAT and PAT–CYS adducts and found that the toxicity of PAT–CYS adducts was significantly less than that of PAT-self, or even without toxicity [25,26]. In recent years, some studies have also reported the reduced toxic effects of thiol-PAT adducts on cells based on the cytotoxicity experiments [15,27,28,29].

As is well-known, the toxicity of the organic compound is closely related to its structure. The toxicity of PAT is mainly decided by the lactone ring, double bonds, and hydroxyl group on its structure [30]. Therefore, the disruption of these critical functional groups and structures could obviously reduce the toxic effects of PAT. As can be seen from Figure 4, the lactone rings on the structures of DP B, D, E, F, G, and H were all destroyed and disappeared. DP A, B, C, F, and G lost the hydroxyl groups on the structure of PAT, while DP A, C, and H changed one or two double bonds to the corresponding single bonds. The toxic effects of these DPs of PAT should be reduced significantly compared to that of PAT-self, while some of them might be still toxic to the tested cells, embryo, animals, and humans, because the critical functional groups and structures of PAT were not completely destroyed. The toxic effects of PAT and its DPs will be furtherly evaluated by the cytotoxicity and animal toxicity experiments and epidemiological investigation in our next study.

## 4. Conclusions

PAT is very stable to heat under highly acidic conditions (pH 3.5), and was very slow to be decomposed in the absence of CYS at 120 °C. In the presence of CYS, high temperatures promoted the degradation of PAT in pH 3.5 of simulated juice solution. A total of eight DPs were identified for the first time according to the accurate mass measurement and calculating ppm error from the LC-Q-TOF-MS analysis. It is very difficult to degrade PAT by CYS without heating treatment based on the Michael addition reaction in a highly acidic environment. The possible reaction mechanisms of PAT degradation by heat-assisted cysteine under the highly acidic conditions were very complicated, and involved the Michael addition reaction, decarboxylation reaction, hydrolysis reaction, esterification reaction, condensation reaction, reduction reaction, thermal dissociation reaction, and so on. The Michael addition reaction is still a critical step in degrading PAT by CYS with the help of heating at 120 °C under highly acidic conditions. The toxic effects of eight DPs were potentially reduced compared to that of PAT-self due to the destruction of critical functional groups and structures (lactone ring, double bond, and –OH) on PAT molecules.

## 5. Materials and Methods

### 5.1. Materials

Standard *L*-cysteine (purity ≥ 99.0%) and PAT (purity ≥ 98.0%) were purchased from the Aladdin Industrial Corporation (Shanghai, China) and the Sangon Biotech (Shanghai, China) Co., Ltd., respectively. Acetonitrile and formic acid were HPLC grade, and were obtained from the Oceanpak Alexative Chemical Ltd., (Gothenburg, Sweden) and the Anpel Laboratory Technologies Inc. (Shanghai, China), respectively. Malic acid and ethyl acetate were analytical grade and purchased from Sinopharm Chemical Reagent Co., Ltd. (Shanghai, China).

### 5.2. Preparation of Simulated Apple Juice Solution (pH 3.5)

In this study, to have a clearer, accurate, and rapid understanding of the DPs and the possible reaction mechanisms of PAT by heat-assisted cysteine under highly acidic conditions and reduce the interference of other substances in apple juice, such as metal ions, vitamin C, and polyphenols, malic acid solution was used to simulated the apple juice. As is well-known, the pH of apple juice is usually around 3.5. To simulate the highly acidic conditions of apple juice, 1 mol/L of malic acid was used to adjust the pH of ultrapure water to 3.5, which was used as the simulated apple juice solution. The pH of the simulated solution was checked with a pH meter (pHs-3C, Leici, Shanghai, China).

### 5.3. Preparation of PAT and Cysteine Solutions with pH 3.5

A PAT solution (0.67 mmol/L or 102.62 mg/L) with pH 3.5 was prepared by diluting 1.0 mg/mL of PAT acetonitrile solution (10.26 mL) to 100 mL in a brown volumetric flask with the simulated apple juice set at pH 3.5. Similarly, the cysteine solution (2.01 mmol/L or 243.53 mg/L) with pH 3.5 was obtained by diluting 1 mg/mL of cysteine aqueous solution (12.18 mL) with the simulated apple juice set at pH 3.5 to 50 mL in a brown volumetric flask.

### 5.4. Determination of PAT in Simulated Apple Juice Solution

The Agilent HPLC 1260 infinity system used for the determination of PAT was composed of quaternary pumps, a column oven, autosampler, and diode array detector (DAD). For the detailed methods, refer to the report by us [16].

### 5.5. Degradation of PAT by Heat-Assisted Cysteine under Highly Acidic Conditions (pH 3.5)

Next, 0.67 mmol/L of PAT solution (pH 3.5) and 2.01 mmol/L of CYS solution (pH 3.5) were rapidly mixed with the same volume (molar ratio of PAT to cysteine was 1:3) and quickly added into a 10-mL kettle. The reaction kettle was placed in an oil bath at 120 °C to react for 30, 60, and 90 min, respectively. Then, the reaction solutions were cooled to room temperature with running water within 3 min. The samples that contained PAT solution (0.67 mmol/L) mixed with the same volume of simulated apple juice (without cysteine) and heated in the same reaction conditions were used as the control.

### 5.6. Structure Analysis of PAT Degradation Products and Possible Reaction Mechanisms

LC-Q-TOF-MS has been widely used to identify the organic compounds and study their fragmentation patterns based on its high sensitivity, large mass range, and simultaneous detection of ions of all masses [20,21,22,23]. In this study, the Agilent LC-Q-TOF-MS system was used to separate and identify the DPs of PAT, which was composed of an Agilent HPLC 1290 infinity system connected to a Q-TOF mass spectrometer (Agilent 6546 series, Palo Alto, CA, USA) through an electron spray ionization (ESI) interface ionization source. Mass Hunter software (B.06.00, Palo Alto, CA, USA) was employed for data processing and acquisition.

LC separation was achieved via an Agilent Infinitylab Poroshell 120 EC-C18 column (100 × 2.1 mm; 2.7 μm) using the ACN:water (5:95, *v/v*) as the mobile phase at 35 °C and a flow rate of 0.3 mL/min. The run time was 30 min with isocratic elution, and the analytes were detected at a wavelength of 276 nm with a DAD detector. One μL of injection volume was sufficient for the separation and identification of PAT and its DPs.

TOF-MS/MS analyses of PAT and its DPs were performed in ESI negative ion mode. The capillary voltage was set at 3500 V, fragmentor voltage at 70 V, and skimmer voltage at 65 V. The drying gas temperature was set at 325 °C with a flow rate of 8.0 L/min, nebulizing gas pressure at 35 psi, and sheath gas temperature at 350 °C with a flow rate of 11.0 L/min. Nitrogen was used as drying, nebulizing, and collision gas.

## Figures and Tables

**Figure 1 toxins-14-00695-f001:**
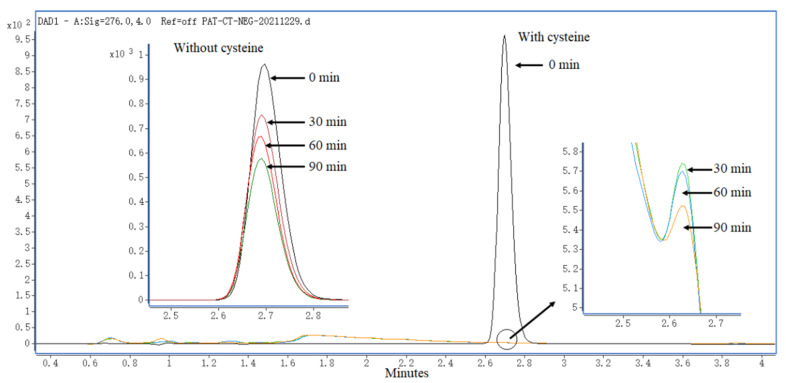
LC-DAD chromatograms of PAT after being degraded by heat-assisted cysteine at 120 °C for different times with or without cysteine.

**Figure 2 toxins-14-00695-f002:**
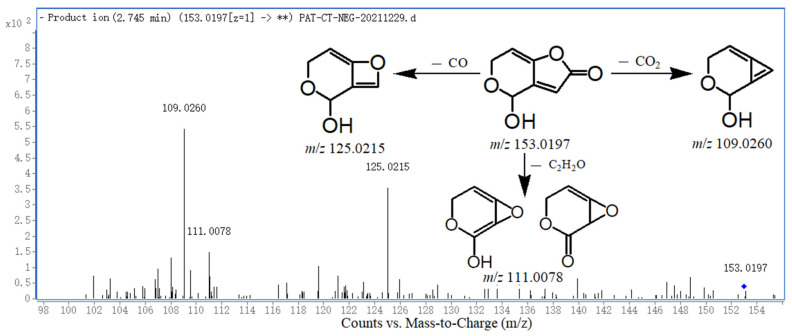
TOF-MS/MS spectrum of PAT and its fragment pattern.

**Figure 3 toxins-14-00695-f003:**
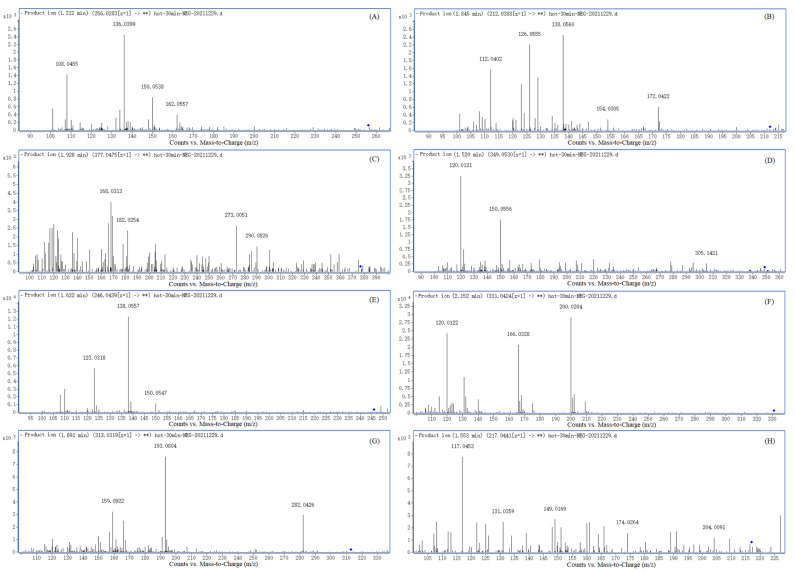
LC-Q-TOF-MS/MS spectra of PAT degradation products (**A**) DP A, (**B**) DP B, (**C**) DP C, (**D**) DP D, (**E**) DP E, (**F**) DP F, (**G**) DP G, and (**H**) DP H. "**"showed that all fragmentation ions of degradation products were listed in the chromatograms.

**Figure 4 toxins-14-00695-f004:**
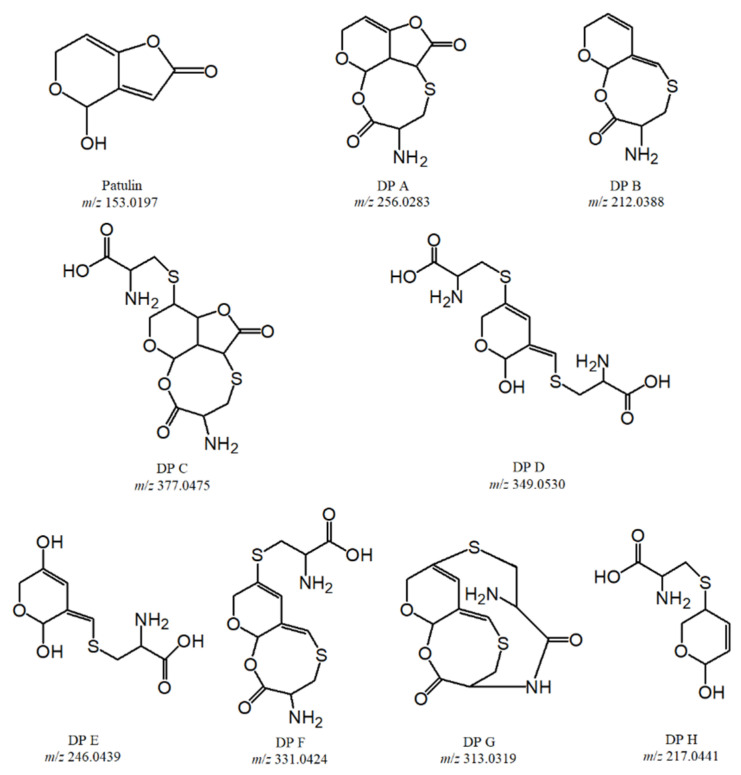
Structures of PAT degradation products (DP A–H) formed by heat-assisted cysteine under highly acidic conditions.

**Figure 5 toxins-14-00695-f005:**
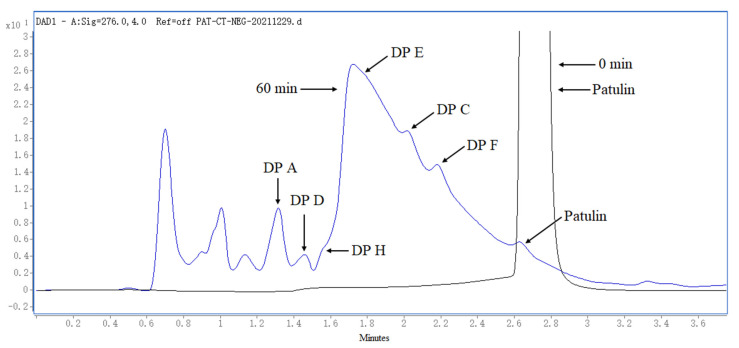
LC-DAD chromatograms of PAT degradation products formed by heat-assisted cysteine under highly acidic conditions.

**Figure 6 toxins-14-00695-f006:**
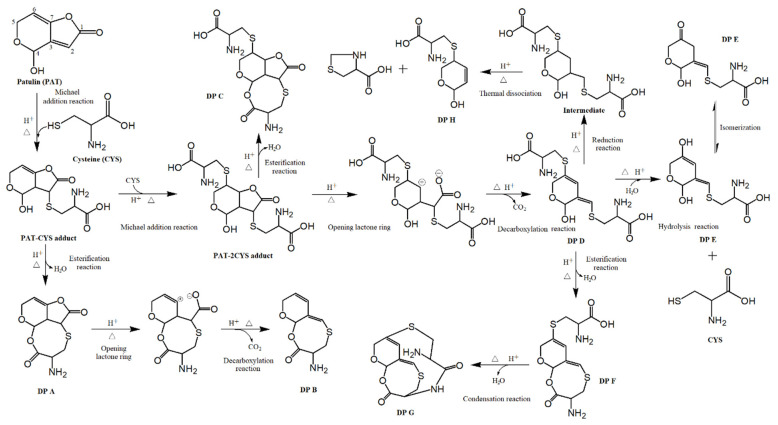
Possible degradation pathway of PAT by heat-assisted cysteine under highly acidic condition.

**Table 1 toxins-14-00695-t001:** LC-Q-TOF-MS/MS data of PAT and its degradation products (DPs) ^a^.

No.	Formula	Retention Time (min)	Observed Mass (*m/z*) ^b^	Calculated Mass (*m/z*) ^b^	Error (ppm)	DBE ^c^	Score ^d^ (%)	Major Fragment Ions
PAT	C_7_H_6_O_4_	2.745	153.0197	153.0193	–2.39	5	99.22	109.0260, 111.0078, 125.0215
DP A ^a^	C_10_H_11_NO_5_S	1.212	256.0283	256.0285	0.84	6	99.81	136.0399, 150.0538, 162.0557
DP B ^a^	C_9_H_11_NO_3_S	1.845	212.0388	212.0387	–0.53	5	100.00	112.0402,126.0560, 138.0560, 154.0305, 172.0422
DP C ^a^	C_13_H_18_N_2_O_7_S_2_	1.928	377.0475	377.0483	2.03	6	98.38	168.0313, 182.0254, 273.0051, 290.0826
DP D ^a^	C_12_H_18_N_2_O_6_S_2_	1.520	349.0530	349.0534	1.00	5	99.63	120.0121, 150.0556, 305.1421
DP E ^a^	C_9_H_13_NO_5_S	1.622	246.0439	246.0442	1.08	4	99.70	123.0318, 138.0557, 150.0556
DP F ^a^	C_12_H_16_N_2_O_5_S_2_	2.152	331.0424	331.0428	1.16	6	99.52	166.0328, 200.0204
DP G ^a^	C_12_H_14_N_2_O_4_S_2_	1.691	313.0319	313.0322	1.03	7	99.65	193.0804, 282.0426
DP H ^a^	C_8_H_12_NO_4_S	1.553	217.0441	217.0441	0.03	4	100.00	117.0453, 131.0359, 174.0264, 204.0091

^a^ DP, the abbreviation of degradation product, and A→H are the numbers of PAT degradation products. ^b^ The mass-to-charge ratio of PAT and its degradation products in negative mode [M–H]^–^. ^c^ DBE, Numbers of the double bond equivalent (i.e., degrees of unsaturation). ^d^ An overall score is 0−100%, with the score closer to 100% being better.

## Data Availability

Not applicable.

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
