# Peer review of "Possible Reaction Mechanisms Involved in Degradation of Patulin by Heat-Assisted Cysteine under Highly Acidic Conditions"

_toxins, 2022, doi:10.3390/toxins14100695_

Round 1

Reviewer 1 Report

The manuscript aimed to deduced the possible reaction mechanisms involved in degradation of patulin by heat-assisted cysteine under highly acidic condition. It was interesting to the readers of journal Toxins and the manuscript was well prepared. I suggest it could be accept after minor revision.

L11-13, L261-264: Authors concluded  the toxic effects of PAT degradation products were far less than those of PAT-self. But I didn't get enough evidence from the data of this manuscript.  I suggest you delete it or supply more supporting data.

L230-248: the food safety is so important that we couldn't ely solely on structural analysis to make judgments. Did you found any previous reports from animal feeding trial or cell culture study?   

Author Response

Dear Reviewer:

  Thank you for the comments concerning our manuscript entitled “Possible reaction mechanisms involved in degradation of Patulin by heat-assisted cysteine under highly acidic condition” (ID: toxins-1825304). Those comments are all valuable and very helpful for revising and improving our paper. We have studied the comments carefully and made corrections which we hope to meet with approval. The main corrections in the paper and the responds to your comments are as follows:

   (1) Comment: The manuscript aimed to deduce the possible reaction mechanisms involved in degradation of patulin by heat-assisted cysteine under highly acidic condition. It was interesting to the readers of journal Toxins and the manuscript was well prepared. I suggest it could be accept after minor revision.

  Response: Thank you very much for your recognition to our study, and we will improve our manuscript according to your suggestions.

  (2) Comment: L11-13, L261-264: Authors concluded  the toxic effects of PAT degradation products were far less than those of PAT-self. But I didn't get enough evidence from the data of this manuscript.  I suggest you delete it or supply more supporting data.

  Response: The question you raised is very good. About the toxic effects of PAT degradation products are deduced based on their molecular structures. Many literatures have reported that the toxicities of chemicals are closely related to their structures (Barratt, 2000; Davidson & Horne, 1983; Pang et al., 2014). In our study, the toxicity of PAT is also closely related to its structure (Wallen et al., 1980). The toxicity of PAT is mainly decided by the lactone ring, double bonds and hydroxyl group on its structure. Therefore, we consider that the disruption of these critical functional groups and structures could obviously reduce the toxic effects of PAT based on the relation between its structure and toxicity.In addition, many literatures also reported that the toxic effects of PAT and PAT-CYS adducts (or thiol-PAT adducts), and found that the toxicity of PAT-CYS adducts was significantly less than that of PAT-self, or even without toxicity (Krivobok et al., 1994; Lindroth &Von Wright, 1990; Liu et al., 2019a; Liu et al.,2019b; Qiu et al., 2020; Rodríguez-Bencomo et al., 2020).According to the reported conclusions, we deduced that the toxic effects of PAT degradation products were far less than those of PAT-self. In order to verify the deduced conclusion, we will do some cytotoxicology, animal toxicology experiments in the next work.

  References

Barratt, M.D. Prediction of toxicity from chemical structure. Cell Biol. Toxicol. 2000, 16(1), 1–13.Davidson, L.N., Horne, R.A. A relationship between toxicity and chemical nature. Physiol. Chem. Phys. Med. NMR. 1983, 15(2), 167–175.

Krivobok, S.; Seigle-Murandi, F.; Sreiman, R.; Benoit-Guyod, J.L.; Bartoli, M.H. Antitumoral activity of patulin and patulin-360 cysteine adducts. Pharmazie 1994, 49(4), 277–279.

Lindroth, S.; Von Wright, A. Detoxification of patulin by adduct formation with cysteine. J. Environ. Pathol. Toxicol. Oncol. 1990, 10(4–5): 254–259.

Liu, M.; Wang, J.; Wang, X.; Zhu, W.; Yao, X.; Su, L.; Sun, J.; Yue, T.; Wang, J. Highly efficient and cost-effective removal of patulin from apple juice by surface engineering of diatomite with sulfur-functionalized graphene oxide. Food Chem. 2019a, 300, 125111. https://doi.org/10.1016/j.foodchem.2019.125111

Liu, M.; Wang, J.; Yang, Q.; Hu, N.; Zhang, W.; Zhu, W.; Wang, R.; Suo, Y.; Wang, J. Patulin removal from apple juice using a  novel cysteine-functionalized metal-organic framework adsorbent. Food Chem. 2019b, 270, 1–9.

Pang, Y.Y., Yeo, W.K., Loh, K.Y., Go, M.L., Ho, H.K. Structure–toxicity relationship and structure–activity relationship study of 2-phenylaminophenylacetic acid derived compounds. Food Chem. Toxicol. 2014, 71, 207–216.

Qiu, Y.; Zhang, Y.; Wei, J.; Gu, Y.; Yue, T.; Yuan, Y. Thiol-functionalized inactivated yeast embedded in agar aerogel for highly efficient adsorption of patulin in apple juice. J. Hazard. Mater. 2020, 388, 121802. https://doi.org/10.1016/j.jhazmat.2019.121802

Wallen, L.L.; Lyons, A.J.; Pridham, T.G. Antimicrobial activity of patulin derivatives: A preliminary report. J. Antibiot. 1980, 33(7), 767–769.

Rodríguez-Bencomo, J.J.; Sanchis, V.; Viñas, I.; Martín-Belloso, O.; Soliva-Fortuny, R. Formation of patulin-glutathione conjugates induced by pulsed light: a tentative strategy for PAT degradation in apple juices. Food Chem. 2020, 315, 126283. https://doi. 350 org/10.1016/j.foodchem.2020.126283.

  (3) Comment: L230-248: the food safety is so important that we couldn't reply solely on structural analysis to make judgments. Did you find any previous reports from animal feeding trial or cell culture study? 

Response: As the reviewer said, food safety is very important and toxicities of PAT and its degradation products cannot be judged only by their structural analysis. We have provided some previous reports to describe their toxicities based on the cell culture study or animal feeding trial (Krivobok et al., 1994; Lindroth &Von Wright, 1990; Liu et al., 2019a; Liu et al.,2019b; Qiu et al., 2020; Rodríguez-Bencomo et al., 2020). About the references, please see the Comment (2).

  Presently, since we have not yet evaluated their toxicities by cell culture study and animal feeding experiments, we only inferred that their toxic effects are reduced based on the relationship between toxicities and structures of PAT and its degradation products.

  Therefore, the 3.3 section in our manuscript, the title is named “3.3. Toxicity Prediction of PAT and its Degradation Products” .

  We tried our best to improve the manuscript and made some changes in the manuscript. These changes will not influence the content and framework of the paper. And here we list the changes. We appreciate for your warm work earnestly, and hope that the correction will meet with approval.

  Once again, thank you very much for your comments and suggestions.

Reviewer 2 Report

The article “Possible reaction mechanisms involved in degradation of patulin by heat-assisted cysteine under highly acidic condition” deals with the detoxification method of patulin in apple juice. As patulin was found to be very stable under the heat treatments and acidic conditions (120℃ and pH 3.5), the cysteine component was added to investigate the patulin degradation. For the first-time eight of patulin degradation products were identified, using LC-MS/MS equipped with TOF mass analyzer. The Michael addition reaction was found to be a critical mechanism in the initial step of patulin degradation, while the following serial reactions facilitated the continuation of the Michael addition reaction. The Authors also deduced that the toxic effects of degradation products are reduced, when compared to the parent molecule, as a result of critical functional groups and structures being destroyed.

The topic of the article corresponds with the aims of the special issue Innovative Mycotoxin Detoxification: Discoveries, Mechanisms and Applications of Toxins, section Mycotoxins. The manuscript seems to be well-structured, providing the sufficient explanation of the research background. The cited references are mainly current, results are clearly presented and discussed, containing publishable data. English language and style are generally satisfying and the manuscript perspective can be easily understood. Please find below a few other minor observations to be addressed before publication.

Line 4/22 …one of mycotoxins usually contaminating apple juice?

Line 8 – Abbreviation explanation for LC-Q-TOF-MS is missing. Also, I would advise to unify abbreviations for the used technique, there is also LC-TOF-MS/MS (Line 61), LC-ESI-Q-TOF/MS (Line 75), LC-Q-TOF/MS (Line 82). As well, please unify LC-DAD (Line 111) and HPLC-DAD (Fig 5).

Line 217 …were increased to 42.12, 78.94, and 100%...Please unify digits (no. of decimal places or significant no.) Also, in Table 1.

Line 290 – …and 2.01 mmol/L of PAT cysteine solution (pH 3.5)?

Line 294 – The samples (do you mean aliquots?) of that PAT solution? This sentence should be rephrased for more clarification.

Line 303 – ESI abbreviation is also without explanation.

Author Response

Dear Reviewer:

  Thank you for the comments concerning our manuscript entitled “Possible reaction mechanisms involved in degradation of Patulin by heat-assisted cysteine under highly acidic condition” (ID: toxins-1825304). Those comments are all valuable and very helpful for revising and improving our paper. We have studied the comments carefully and made corrections which we hope to meet with approval. The main corrections in the paper and the responds to the reviewers’ comments are as follows:

(1) Comment: Line 4/22 – …one of mycotoxins usually contaminating apple juice?

Response: Thank you for pointing us out the grammatical errors. According to your suggestion, we have revised them as follows:

Line 4: Patulin (PAT) is one of mycotoxins that usually contaminates apple juice.

Line 22: PAT is one of mycotoxins that usually contaminates apple juice.

(2) Comment: Line 8 – Abbreviation explanation for LC-Q-TOF-MS is missing. Also, I would advise to unify abbreviations for the used technique, there is also LC-TOF-MS/MS (Line 61), LC-ESI-Q-TOF/MS (Line 75), LC-Q-TOF/MS (Line 82). As well, please unify LC-DAD (Line 111) and HPLC-DAD (Fig 5).

Response: Thanks you for the good idea and suggestions. We supply the full names of these techniques where they first appear in the manuscript, and unify them.

Line 8: liquid chromatography quadrupole time-of-flight mass spectrometer (LC-Q-TOF-MS)

Line 10: MS/MS analysis: Mass spectrometry/mass spectrometry (MS/MS), or tandem mass spectrometry, is a procedure for improving the specificity of the mass spectrometer. When two mass analysers are coupled with each other by using a collision cell we call them as MS/MS or in tandem MS.

Line 61: liquid chromatography quadrupole time-of-flight mass spectrometer/mass spectrometer (LC-Q-TOF-MS/MS).

Line 75: “LC-ESI-Q-TOF/MS” has been changed to “LC-Q-TOF-MS”.

Line 82: “LC-Q-TOF/MS” has been changed to “LC-Q-TOF-MS”.

Line 111: liquid chromatography coupled with diode array detector (LC-DAD).

Line 123: “HPLC-DAD” has been changed to “LC-DAD”.

Line 303: electron spray ionization (ESI).

(3) Comment: Line 217 – …were increased to 42.12, 78.94, and 100%...Please unify digits (no. of decimal places or significant no.) Also, in Table 1.

Response: Thanks you for the good idea and suggestions. We have unified the digits in our manuscript, including the digits in the Table 1.

(4) Comment: Line 290 – …and 2.01 mmol/L of PAT cysteine solution (pH 3.5)?

Response: Thank you very much for pointing out this writing error due to our carelessness. We have corrected this error.

  “0.67 mmol/L of PAT solution (pH 3.5) and 2.01 mmol/L of PAT solution (pH 3.5) were rapidly mixed with the same volume……” has been revised to “0.67 mmol/L of PAT solution (pH 3.5) and 2.01 mmol/L of CYS solution (pH 3.5) were rapidly mixed with the same volume……”

(5) Comment: Line 294 – The samples (do you mean aliquots?) of that PAT solution? This sentence should be rephrased for more clarification.

Response: Thank you very much for pointing out the fuzzy expression. According to your suggestion, we rephrased it for more clarification.

  The samples that contained PAT solution (0.67 mmol/L) mixed with the same volume of simulated apple juice (without cysteine) and heated in the same reaction conditions were used as the control.

(6) Comment: Line 303 – ESI abbreviation is also without explanation.

Response: Based on your suggestion, we suppled the full name of ESI.

  Line 303: electron spray ionization (ESI).

  We tried our best to improve the manuscript and made some changes in the manuscript. These changes will not influence the content and framework of the paper. And here we list the changes. We appreciate for your warm work earnestly, and hope that the correction will meet with approval.

  Once again, thank you very much for your comments and suggestions.

Reviewer 3 Report

The manuscript is not ready for publication. There are various issues related to study designs and data interpretation. Below are some examples.

1. Figure 1. The presentation here is misleading because no validated quantitative method was used to monitor the change of the concentration of patulin. The authors should have validated the LC method in terms recovery, linear range, reproducibility...

2. The concentration of cysteine should have been monitored as well as the authors claimed that cysteine reacted with patulin in the course of heating.

3. Did the authors evaluate effects of temp. and concentrations of cysteine? The Data are missing. It is unclear why 120oC and  1:3 ratio (patulin to cysteine) were chosen.

4. The author used some concentrations of patulin at 0.67 mmol/L and 2.01 mmol/L, which are much higher than established regulatory levels (10-50 ppb). After 90 min at 120oC with/without cysteine, what were the residual concentrations of patulin? The information ought to be provided.

5. Figure 5. Without sufficient separation one cannot claim those "peaks" are degradation products of patulin. 

6. Table 1. Clarify how score values were generated. 

7. There are no clear connections between the products reported in 2.3.1-2.3.8 and patulin.

8. Unless the degradation experiments are performed in apple juice, the proposed degradation pathways/products are meaningless.    

Author Response

Dear Reviewer:

  Thank you for the comments concerning our manuscript entitled “Possible reaction mechanisms involved in degradation of Patulin by heat-assisted cysteine under highly acidic condition” (ID: toxins-1825304). Those comments are all valuable and very helpful for revising and improving our paper. We have studied the comments carefully and made corrections which we hope to meet with approval. The main corrections in the paper and the responds to the reviewers’ comments are as follows:

(1) Comment: Figure 1. The presentation here is misleading because no validated quantitative method was used to monitor the change of the concentration of patulin. The authors should have validated the LC method in terms recovery, linear range, reproducibility...

Response: First of all, thank you very much for pointing out this problem that we took into account before writing this manuscript. In this manuscript, Figure 1 is only used to show the degradation effects of patulin in simulated apple juice (pH 3.5) by heating at 120℃ with and without the presence of cysteine. The LC-DAD chromatograms more intuitively show the concentration change of patulin in the presence of cysteine and absence of cysteine after being heated treatment at 120℃ for different times (0~90 min). In addition, we have provided the Table S1 in the Supplementary Materials, which shows the Residual patulin and its degradation rate after being treated by heating at 120℃ for different times with or without cysteine.

Table S1 Residual patulin and its degradation rate after being treated by heating at 120℃ for different times with or without cysteine

Reaction Time

(min)

Residual patulin (mg/L)

Degradation rate of patulin (%)

Without cysteine

With cysteine

Without cysteine

With cysteine

0

51.31

51.31

0.00

0.00

30

41.46

0.05

19.19

99.90

60

36.73

0.05

28.42

99.91

90

31.59

0.04

38.43

99.93

About the determination method of patulin, we have described it at “5.4. Determination of PAT in Simulated Apple Juice Solution” of this manuscript.

(2) Comment: The concentration of cysteine should have been monitored as well as the authors claimed that cysteine reacted with patulin in the course of heating.

Response:

In this study, we focused on exploring the possible reaction mechanisms involved in degradation of patulin by heat-assisted cysteine under highly acidic condition. According to the Michael addition reaction, 1 molar of patulin can completely react with 2 molar of cysteine to form PAT-2CYS adducts. To ensure thorough degradation of patulin, cysteine was excess in this experiment. Therefore, we set the molar ration of patulin to cysteine at 1:3.

Based on the above considerations, we only determined the concentration changes of patulin by heat-assisted cysteine under highly acidic condition. It does not affect the purpose of this study, which is to identify the degradation products of patulin by the LC-Q-TOF-MS/MS analysis, and deduced the possible reaction mechanisms involved in the degradation of patulin by heat-assisted cysteine under the highly acidic condition (pH 3.5).

 (3) Comment: Did the authors evaluate effects of temp. and concentrations of cysteine? The Data are missing. It is unclear why 120℃ and  1:3 ratio (patulin to cysteine) were chosen.

Response: About the factors influencing patulin degradation by heating treatment in highly acidic conditions, we have been studied and published in Toxins (Diao et al., 2021). In our reported literature, we investigated the effects of reaction temperature, reaction time, with or without cysteine on patulin degradation by thermal treatment in highly acidic conditions. Based on our previous study, we found that patulin in highly acidic conditions could be effectively degraded by thermal treatment at 120℃ for different times (30~180 min) in the presence of cysteine (Diao et al., 2021). So, in this study, we chose the reaction temperature of 120℃ to explore the possible reaction mechanisms involved in degradation of patulin by heat-assisted cysteine under highly acidic condition.

According to the Michael addition reaction, 1 molar of patulin can completely react with 2 molar of cysteine to form PAT-2CYS adducts. To ensure thorough degradation of patulin, cysteine was excess in this experiment. Therefore, we set the molar ration of patulin to cysteine at 1:3.

References

Diao, E.J., Ma, K., Zhang, H., Xie, P., Qian, S.Q., Song, H.W., Mao, R.F., Zhang, L.M. Thermal stability and degradation kinetics of patulin in highly acidic conditions: Impact of cysteine. Toxins, 2021, 13, 662. Https://doi.org/10.3390/toxins13090662.

(4) Comment: The author used some concentrations of patulin at 0.67 mmol/L, which is much higher than established regulatory levels (10-50 ppb). After 90 min at 120℃ with/without cysteine, what were the residual concentrations of patulin? The information ought to be provided.

Response: As the reviewer said, many international organizations (such as WHO, EU) and countries have established regulatory levels at 10-50 ppb in apple-based products. 0.67 mmol/L (Equivalent to 103.26 mg/L or 103.26 ppm) of patulin was chosen to investigate its degradation mechanisms by heat-assisted cysteine under highly acidic condition, which was much higher than established regulatory levels (10-50 ppb). In this study, 0.67 mmol/L of patulin (pH 3.5) was rapidly mixed with the same volume of cysteine (pH 3.5), so the final concentration of patulin in reaction solution was 0.335 mmol/L (Equivalent to 51.63 mg/L or 51.63 ppm, the actual concentration of patulin is 51.31 mg/L or 51.31 ppm based on the LC-DAD analysis, See Table S1). This concentration of patulin is approximately 1000 times of regulatory levels (50 ppb).

Why do we choose this high concentration of patulin?

From the perspective of food safety, the concentration of 50 ppb is very high, which reaches the regulatory levels set by WHO and EU. However, this concentration is still relatively low based on the accurate analysis. Especially, the concentrations of the degradation products of patulin will be very low when the initial concentration of patulin is less than 50 ppb, which can not be detected effectively by LC-Q-TOF-MS. Therefore, for a clearer isolation and more accurate identification of the patulin degradation products, we increased its concentration to 1000 times of the regulatory levels (50 ppb).

Overall, the higher concentration of patulin was solely designed to facilitate the isolation and identification of its degradation products, and then to deduce the possible reaction mechanisms of between patulin and cysteine in highly acidic conditions in this study.

The residual concentrations of patulin in highly acidic condition after being treated at 120℃ for different times (0~90min) have been provided in Table S1 in the Supplementary Materials. In this study, our main objective is to isolate and identify the degradation products of patulin by the LC-Q-TOF-MS/MS analysis, and then to deduce the possible reaction mechanisms involved in the degradation of patulin by heat-assisted cysteine under the highly acidic condition (pH 3.5). So the degradation efficiency of patulin is not the focus, and it is only described simply in this study. The detailed data were provided in the Supplementary Materials considering the repeatability of our published results (Diao et al., 2021) and the limitations on the numbers of figures and tables in this manuscript.

(5) Comment: Figure 5. Without sufficient separation one cannot claim those "peaks" are degradation products of patulin.

Response: Figure 5 shows the LC-DAD chromatograms of patulin degradation products formed by heat-assisted cysteine at 120℃for 0 min (the control) and 60 min under highly acidic condition, respectively. Seen from Figure 5, six new products were formed during heating treatment of 60 min, as compared to the control (0 min).

As the reviewer said, the degradation products of patulin were not separated sufficiently, which could not affect the identification of these products. Because LC-Q-TOF-MS has been widely used to identify the organic compounds and study their fragmentation patterns based on its high sensitivity, large mass-range, and simultaneous detection of ions of all masses (Pokar, Sahu, & Sengupta, 2020; Velip et al., 2022; Wan et al., 2021; Zhao et al., 2022; Zhou et al., 2017). Therefore, in this study, a total of eight degradation products (DP A–H) were characterized and identified by LC-Q-TOF-MS in a negative ion mode, and their structures and formulas were proposed based on their accurate mass data.

References

Pokar, D.; Sahu, A.K.; Sengupta, P. LC-Q-TOF-MS driven identification of potential degradation impurities of venetoclax, mechanistic explanation on degradation pathway and establishment of a quantitative analytical assay method. J. Anal. Sci. Technol. 2020, 11, 54. https://doi.org/10.1186/s40543-020-00252-4

Velip, L.; Dhiman, V.; Kushwah, B.S.; Golla, V.M.; Gananadhamu, S. Identification and characterization of urapidil stress degradation products by LC-Q-TOF-MS and NMR: Toxicity prediction of degradation products. J. Pharmaceut. Biomed. 2022, 211, 114612. https://doi.org/10.1016/j.jpba.2022.114612

Wan, J.; He, P.; Chen, Y.S.; Zhu, Q.J. Comprehensive target analysis for 19 pyrethroids in tea and orange samples based on LC-ESI-QqQ-MS/MS and LC-ESI-Q-ToF/MS. LWT 2021, 151, 112072. https://doi.org/10.1016/j.lwt.2021.112072

Zhao, R.X.; Zou, H.; Zhao, R.J.; Li, N.Y.; Zheng, Z.J.; Qiao, X.G. Effect of amino acids on formation of pigment precursors in garlic discoloration using UPLC-ESI-Q-TOF-MS analysis. J. Food Compos. Anal. 2022, 105, 104231. https://doi.org/10.1016/j.jfca.2021.104231

Zhou, N.; Qian, Q.; Qi, P.C.; Zhao, J.; Wang, C.Y.; Wang, Q. Identification of degradation products and process impurities from terbutaline sulfate by UHPLC-Q-TOF-MS/MS and in silico toxicity prediction. Chromatographia 2017, 80, 793–804.

(6) Comment: Table 1. Clarify how score values were generated. 

Response: The score values are molecular formula generation (MFG) matching scores, which include MS level scoring and MS/MS level scoring. They were obtained from the Molecular Calculator (Figure 1) in the Mass Hunter software (B.06.00). The score is ranged from 0 to 100%, with the score closer to 100% being better.

MS Level Scoring: Takes into account mass accuracy (MS), isotopic abundance and isotopic spacing.

MS/MS Level Scoring: Takes into account mass accuracy (MS/MS) of fragment ions and neutral loss information.

MFG Score, Overall = MFG Score (MS) + MFG Score (MS/MS ).

Figure 1 Data information from the Molecular Calculator (Figure 1) in the Mass Hunter software (B.06.00) (Patulin as an example)  

(7) Comment: There are no clear connections between the products reported in 2.3.1-2.3.8 and patulin.

Response: In this study, from 2.3.1 to 2.3.8, we identified the structures of degradation products A-H of patulin based on their Q-TOF-MS/MS spectra. Figure 4 shows their molecular structures in our manuscript. Seen from the Figure 4, the structures of the degradation products A-H were closely related to the ones of patulin and cysteine, which were adducts of patulin and cysteine, such as PAT-CYS adducts, PAT-2CYS adducts. Their formation processes were very complex, which involved the Michael addition reaction, decarboxylation reaction, hydrolysis reaction, esterification reaction, condensation reaction, reduction reaction, and thermal dissociation reaction, and so on. The Michael addition reaction is still a critical step in degrading PAT by CYS with the help of heating at 120℃ under the highly acidic condition.

Therefore, we considered that There are clear connections between the products reported in 2.3.1-2.3.8 and patulin.

(8) Comment: Unless the degradation experiments are performed in apple juice, the proposed degradation pathways/products are meaningless.

Response: In this study, our main objective is to isolate and identify the degradation products of patulin by the LC-Q-TOF-MS/MS analysis, and then to deduce the possible reaction mechanisms involved in the degradation of patulin by heat-assisted cysteine under the highly acidic condition (pH 3.5). In order to have a clearer, accurate and rapid understanding of the degradation mechanisms of patulin by heat-assisted cysteine under highly acidic condition, and reduce the interference of other substances (such as apple juice, which contains nutritional components, metal ions, vitamins, and polyphenols, and so on), we used the patulin (purity≥98.0%) and cysteine (purity≥99.0%) in simulated apple juice (pH 3.5) to react at the set conditions, and then to isolate and identify the degradation products of patulin by the LC-Q-TOF-MS/MS analysis. Therefore, we believe that our experimental design scheme is unproblematic. And this scheme can be more accurate and clear to obtain our experimental objective.

  We tried our best to improve the manuscript and made some changes in the manuscript. These changes will not influence the content and framework of the paper. And here we list the changes. We appreciate for your warm work earnestly, and hope that the correction will meet with approval.

  Once again, thank you very much for your comments and suggestions.

Reviewer 4 Report

The manuscript is an interesting work about patulin degradation by heat-assisted cysteine under acidic conditions. The work reports a series of putative “degradation” products that result from the reaction of patulin with cysteine. I think authors should consider if “degradation products” is the best way to define these compounds. It seems they are more “transformation products”, as they all possess more MW than patulin due to cysteine adduction. The abstract and introduction need some English improvement in some passages. I leave other comments in the attached document for correction. Regarding the toxicity of these compounds, I think authors should avoid saying they are less toxic than patulin. Better to express they are potentially less toxic due to the reasons mentioned in the manuscript. This is critical in the abstract. Please revise.

Author Response

Dear Reviewer:

  Thank you for the comments concerning our manuscript entitled “Possible reaction mechanisms involved in degradation of Patulin by heat-assisted cysteine under highly acidic condition” (ID: toxins-1825304). Those comments are all valuable and very helpful for revising and improving our paper. We have studied the comments carefully and made corrections which we hope to meet with approval. The main corrections in the paper and the responds to the reviewers’ comments are as follows:

(1) Comment: I think authors should consider if “degradation products” is the best way to define these compounds. It seems they are more “transformation products”, as they all possess more MW than patulin due to cysteine adduction.

Response: Thank you very much for pointing out this question. According to our investigation, many methods are used to degrade or remove mycotoxins (such as aflatoxins, patulin, and so on) from foods and agricultural products, which include physical, chemical, and biological methods (Afsah-Hejri, Hajeb, Ehsani, 2020; Diao et al., 2018; Ji, Fan, & Zhao, 2016). According to the above-mentioned reviews, these methods can effectively degrade or remove mycotoxins. And the word “degrade” is frequently used to describe the reduction or disappearance of mycotoxins after being treated by physical, chemical, or biological methods. In this study, heat-assisted cysteine is used to reduce patulin in highly acidic condition, which belongs to a combination of physical and chemical methods. Therefore, we continue to use the word “degradation” to describe the reduction of patulin in this study.

  Seen from the Figure 4 in our manuscript, the structures of the degradation products A-H were closely related to the ones of patulin and cysteine, which were adducts of patulin and cysteine, such as PAT-CYS adducts, PAT-2CYS adducts. So these products possess more MW than patulin due to cysteine adduction. Their formation processes were very complex, which involved the Michael addition reaction, decarboxylation reaction, hydrolysis reaction, esterification reaction, condensation reaction, reduction reaction, and thermal dissociation reaction, and so on. The Michael addition reaction is still a critical step in degrading PAT by CYS with the help of heating at 120℃ under the highly acidic condition.

  According to the above-mentioned analysis, the possible reaction mechanisms involved in degradation of patulin contained a series of complex reaction processes such as addition reaction and decomposition reaction, and resulted in the molecular weight of patulin degradation products were all greater than that of patulin-self.

References

Ji, C., Fan, Y., Zhao, L.H. Review on biological degradation of mycotoxins. Animal Nutrition, 2016, 2(3), 127–133.

Afsah-Hejri, L., Hajeb, P., Ehsani, R.J. Application of ozone for degradation of mycotoxins in food: A review. Comprehensive Reviews in Food Science and Food Safety, 2020, 19(4), 1777–1808.

Diao, E.J., Hou, H.X., Hu, W.C., Dong, H.Z., Li, X.Y. Removing and detoxifying methods of patulin: A review. Trends in Food Science & Technology, 2018, 81, 139–145.

(2) Comment: The abstract and introduction need some English improvement in some passages.

Response: Thanks the reviewer for pointing out the grammar errors, and we improved the full English with your suggestions.

Line 4: Patulin (PAT) is one of mycotoxins that usually contaminates apple juice,…..

Line 12-13: it could be deduced that the toxic effects of PAT degradation products were potentially much less than those of PAT-self.

Line 14: About the Keywords, we have deleted their abbreviations.

Line 22: Patulin (PAT) is one of mycotoxins that usually contaminates apple juices, ……

Line 29: According to our investigation, PAT in apple juice is removed mainly by food processing……

Line 33-36: Presently, other physical, chemical or biological methods for detoxifying PAT have not been adopted for commercial application in juice industry due to some drawbacks, such as the lower detoxification efficiency, higher cost in detoxifying PAT, larger damage to the nutrition and sensory quality of juice, easily causing cross-contamination to juice and environment by undestroyed toxin, and unrealized industrial application due to the limited conditions.

Line 40: ……and even caused the cross-contamination by it due to the improper operation during processing.

(3) Comment: I leave other comments in the attached document for correction.

Response: Thank you very much for your suggestions. We have tried to revise and improve the corresponding problems based on your suggestions.

We changed the upper case letters in the titles of all figures and tables to lower case ones

Line 73: Figure 1. LC-DAD chromatograms of PAT after being degraded by heat-assisted cysteine at 120℃ for different times with or without cysteine.

Line 95: Figure 2. TOF-MS/MS spectrum of PAT and its fragment pattern.

Line 97: Table 1. LC-Q-TOF-MS/MS data of PAT and its degradation products (DPs).

Line 117-118: Figure 3. LC-Q-TOF-MS/MS spectra of PAT degradation products (A) DP A, (B) DP B, (C) DP C, (D) DP D, (E) DP E, (F) DP F, (G) DP G, and (H) DP H.

Line 120: Figure 4. Structures of PAT degradation products (DP A–H) formed by heat-assisted cysteine under highly acidic condition.

Line 123-124: Figure 5. LC-DAD chromatograms of PAT degradation products formed by heat-assisted cysteine under highly acidic condition.

Line 201: Figure 6. Possible degradation pathway of PAT by heat-assisted cysteine under highly acidic condition.

Line 282: ……the simulated apple juice set at pH 3.5.

Line 283-284: Similarly, the cysteine solution (2.01 mmol/L or 243.53 mg/L) with pH 3.5 was obtained by diluting 1 mg/mL of cysteine aqueous solution (12.18 mL) with the simulated apple juice set at pH 3.5 to 50 mL in a brown volumetric flask.

Line 294: “……with the running water…….” Changes to “…… with running water…….”.

Line 294-296: The samples that contained PAT solution (0.67 mmol/L) mixed with the same volume of simulated apple juice (without cysteine) and heated in the same reaction conditions were used as the control.

(4) Comment: Regarding the toxicity of these compounds, I think authors should avoid saying they are less toxic than patulin. Better to express they are potentially less toxic due to the reasons mentioned in the manuscript. This is critical in the abstract. Please revise.

Response: Thank you very much for your good suggestion. According to your suggestion, we improved the language description for more objective expression.

Line 262: The toxic effects of eight DPs were potentially reduced compared to that of PAT-self……

  We tried our best to improve the manuscript and made some changes in the manuscript. These changes will not influence the content and framework of the paper. And here we list the changes. We appreciate for your warm work earnestly, and hope that the correction will meet with approval.

  Once again, thank you very much for your comments and suggestions.

Round 2

Reviewer 3 Report

The authors have only made a few cosmetic changes in the revised version so I cannot recommend the manuscript for publication. There are various issues in authors' responses. Below please find a few examples.

Response 1. The authors have failed to provide method validation data to demonstrate the performance (e.g., linear range, sensitivity, recovery, reproducibility...) of the LC method used to monitor concentrations of patulin. Citing a published paper (Diao, et al., 2021) is not helping. As in that paper, required key info is still missing. Basically, the authors have been using a method that has not been validated for quantitative analysis in different studies.

Response 2. Without monitoring cysteine, how could the authors be sure cysteine is involved in the degradation of patulin, especially the generation of reported degradation products?     

Response 5. Even in the revised version, the authors still label those un resolved "peaks" as identified degradation products. Unless the authors could use one of the peaks as the example to explain how they achieved identification of the according compound, it is still unclear how identification was achieved. Without reference standards available or NMR data, it would require more efforts to say the peak at 1.5 min is proposed DP E. Furthermore, there is no clear connection between patulin and DP E or any proposed degradation products. It is amazing to see the authors could come up the structure of DP G.  

Response 8. The authors have claimed that cysteine can be used to eliminate patulin while refused to test the notion in the real-life situation, showing subjective bias to what should be a more objective look at what is being done. Furthermore, the lack of the data in apple juice suggests such approach has not practical benefits.

Therefore, I cannot agree to publish this article in its current form for these reasons and many more I could elaborate but feel no need to do so now if the others feel differently anyway.

Author Response

Dear Reviewer:

Thank you so much for giving us the opportunity to revise our manuscript again. We have carefully considered your questions and comments about our manuscript, and made the following responses.

(1) Comment: The authors have failed to provide method validation data to demonstrate the performance (e.g., linear range, sensitivity, recovery, reproducibility...) of the LC method used to monitor concentrations of patulin. Citing a published paper (Diao, et al., 2021) is not helping. As in that paper, required key info is still missing. Basically, the authors have used a method that has not been validated for quantitative analysis in different studies.

Response: Thank you very much for this question. As a paper focusing on the development of detection methods, the validation data to demonstrate the performance (e.g., linear range, sensitivity, recovery, reproducibility...) of the LC method must be provided. In this study, the main objective of this study is to identify the DPs of PAT by LC-Q-TOF-MS/MS analysis, and deduced the possible reaction mechanisms involved in the degradation of PAT by heat-assisted cysteine under the highly acidic condition (pH 3.5). So the detection of patulin by HPLC-DAD method is simply to determine whether patulin was degraded by heat-assisted cysteine, which provides support for the next separation and recognition of the patulin degradation products.

In addition, HPLC coupled with ultraviolet (UV) or photodiodes (DAD) detection is the most used chromatographic technique for patulin analysis; it allows an easy identification and quantification of patulin through its characteristic absorption spectrum. The AOAC also adopted HPLC-UV method (official method 995.10), for the detection and quantification of patulin. In this study, we used the LC-DAD method to determine the concentrations of patulin, which is very mature, and has been widely used in the detection of patulin by many researchers (Brause, 1996; Forbito & Babsky, 1996; Katerere, Stockenstron, & Shephard, 2008; MacDond et al., 2000; Moukas, Panagiotopoulou, & Markaki, 2008; Sadok, Szmagara, & Staniszewska, 2017; Sadok, Stachniuk, & Staniszewska, 2019). Therefore, we believe that LC-DAD method used in our study is a reliable and accurate method for determining patulin.

References

AOAC Official Method 995.10. Patulin in Apple juice, Liquid Chromatographic Method. https://img.21food.cn/img/biaozhun/20100108/177/11285242.pdf

Brause, A.R. Determination of patulin in apple juice by liquid chromatography: collaborative study. Journal of AOAC International, 1996, 79(2), 451–455.

Forbito, P.R., Babsky, N.E. Rapid liquid chromatographic determination of patulin in apple cider. Journal of Chromatography, 1996, 730, 53–58.

Katerere, D.R., Stockenstron, S., Shephard, G.S. HPLC-DAD method for the determination of patulin in dried apple rings. Food Control, 2008, 19(4), 389–392.

MacDond, S., Long, M., Gilbert, J., Felgueiras, I. Liquid chromatographic method for determination of patulin in clear and cloudy apple juices and apple puree: collaborative study. Journal of AOAC International, 2000, 83(6), 1387–1394.

Moukas, A., Panagiotopoulou, V., Markaki, P. Determination of patulin in fruit juices using HPLC-DAD and GC-MSD techniques. Food Chemistry, 2008, 109(4), 860–867.

  Sadok, I., Szmagara, A., Staniszewska, M.M. The validated and sensitive HPLC-DAD methods for determination of patulin in strawberries. Food Chemistry, 2017, 245, 364–370.

Sadok, I., Stachniuk, A., Staniszewska, M.M. Developments in monitoring of patulin in fruits using liquid chromatography: an overview. Food Analytical Methods, 2019, 12, 76–93.

(2) Comment: Without monitoring cysteine, how could the authors be sure cysteine is involved in the degradation of patulin, especially the generation of reported degradation products?

Response: In order to effectively degrade patulin, we added excess cysteine, i.e. The molar ration of patulin to cysteine is 1:3 (theoretically, 1 molar of the patulin can react with 2 moles of the cysteine to form PAT-2CYS conduct based on the Michael addition reaction under near neutral conditions). In our reported literature, we found that cysteine could effectively degrade or reduce patulin in pH 3.5 of simulated solution (Diao et al., 2021). Meanwhile, our team also found that cysteine also promotes ultrasonic degradation of patulin in highly acidic condition (Ma et al., 2022).

  To confirm that cysteine was involved in the patulin degradation in highly acidic condition, we performed the comparison experiments (Figures 1 and S1). Seen from the Figure 1 and S1, patulin in pH 3.5 of the simulated juice solution was only reduced by 19.19~38.43% after heating treatment at 120℃ for the times ranged from 30 to 90 min in the absence of cysteine. In the presence of cysteine, patulin was almost completely degraded, and the degradation rate was greater than 99.9% after heating treatment at 120℃ for 30 min (Figures 1 and S1).

Figure 1 LC-DAD chromatograms of PAT after being degraded by heat-assisted cysteine at 120℃ for different times with or without cysteine.

Figure S1 Residual patulin and its degradation rate after being treated by heating at 120℃ for different times with or without cysteine

  Based on the above analysis, we concluded that the cysteine was involved in the patulin degradation. About the degradation products of patulin, we also isolated and identified them by LC-Q-TOF-MS/MS analysis.

References

Diao, E.J., Ma, K., Zhang, H., Xie, P., Qian, S.Q., Song, H.W., Mao, R.F., Zhang, L.M. Thermal stability and degradation kinetics of patulin in highly acidic conditions: impact of cysteine. Toxins, 2021, 13, 662. https://doi.org/10.3390/toxins13090662.

Ma, K., Zhang, H., Diao, E.J., Qian, S.Q., Xie, P., Mao, R.F., Song, H.W., Zhang, L.M. Cysteine-enhanced ultrasound degradation of patulin in acidic solution simulated pH of apple juice. Journal of Food Processing and Preservation, 2022, 00, e16547. https://doi.org/10.1111/jfpp.16547.

(3) Comment: Even in the revised version, the authors still label those unresolved "peaks" as identified degradation products. Unless the authors could use one of the peaks as the example to explain how they achieved identification of the according compound, it is still unclear how identification was achieved. Without reference standards available or NMR data, it would require more efforts to say the peak at 1.5 min is proposed DP E. Furthermore, there is no clear connection between patulin and DP E or any proposed degradation products. It is amazing to see the authors could come up the structure of DP G.  

Response: The reviewer may not be clear about the process of identifying the structures of unknown substances using LC-Q-TOF-MS/MS. Here, we will show you how we identify the structure of unknown products by LC-Q-TOF-MS/MS.

Firstly, we used the liquid chromatography to isolated the patulin degradation products formed by heat-assisted cysteine under highly acidic condition, and the sample that contained patulin solution in the absence of cysteine was used as the control (Figure 5 in our manuscript, the blue line represents the chromatogram containing degradation products of patulin, and black line represents the control sample). Seen from the Figure 5, the control sample only showed one chromatographic peak at 2.745 min, while more than ten chromatographic peaks appeared in the sample treated by heat-assisted cysteine. Several chromatographic peaks were not isolated completely, but it did not affect the analysis of the primary and secondary MS of these unknown substances due to the high sensitivity, large mass-range, and simultaneous detection of ions of all masses for LC-Q-TOF-MS/MS.

Figure 5. LC-DAD chromatograms of PAT degradation products formed by heat-assisted cysteine under highly acidic condition.

Secondly, according to the LC-DAD chromatograms, we performed the primary MS analysis of all the new peaks (possibly product peaks) present in the tested sample based on the retention time of each peak in the ESI negative ion mode by Q-TOF-MS, and then obtained their parent ions.

Thirdly, the parent ions continued to be fragmented in ESI negative ion mode according to the parent ion masses and their retention times to obtain the daughter ions (or fragment ions) by Q-TOF-MS/MS. The parent ion structures were inferred by the fragment patterns of the parent ion, and the daughter ion masses (Figure 2 in our manuscript). In our manuscript, “2.2. Fragment Patterns of PAT Based on the LC-Q-TOF-MS Analysis” as an example, has shown the parent ion, the daughter ions of patulin in the ESI negative ion mode by Q-TOF-MS/MS and the structure identification process of patulin.

The identification of degradation products of patulin sees the “2.3. Identification of PAT Degradation Products Based on the LC-Q-TOF-MS Analysis” in our manuscript.

Figure 2. TOF-MS/MS spectrum of PAT and its fragment pattern.

In this study, a total of eight DPs (DP A–H) were identified and characterized based on the accurate mass, error ppm, and DBE (Figure 4 in our manuscript). Seen from the Figure 4, the structures of these eight products are PAT-CYS or PAT-2CYS adducts based on the Michael addition reaction, or some products from PAT-CYS or PAT-2CYS adducts by the decarboxylation reaction, hydrolysis reaction, esterification reaction, condensation reaction, reduction reaction, and thermal dissociation reaction, and so on. These product structures are closely related to the ones of patulin and cysteine (Figure 4).

Figure 4. Structures of PAT degradation products (DP A–H) formed by heat-assisted cysteine under highly acidic condition.

The degradation products of patulin (DP A-H) were named based on their orders in which they appeared in the possible degradation pathway of patulin (Figure 6 in our manuscript). The parent ions and fragment ions of patulin degradation products have the fixed retention times and accurate m/z ratio, so we can clearly identify the different degradation products based on the chromatographic peaks.

Figure 6. Possible degradation pathway of PAT by heat-assisted cysteine under highly acidic condition.

As the reviewer said, it is better to verify the structures of patulin degradation products by NMR. In the first response to the comment, the reviewer pointed out that the concentration of 50 ppb is very high from the perspective of food safety, which reaches the regulatory levels set by WHO and EU. However, this concentration is still relatively low based on the accurate analysis. Especially, the concentrations of the degradation products of patulin will be very low when the initial concentration of patulin is less than 50 ppb, which can not be detected effectively by LC-Q-TOF-MS. Therefore, for a clearer isolation and more accurate identification of the patulin degradation products, we increased its concentration to 1000 times of the regulatory levels (50 ppb), which can be detected by LC-Q-TOF-MS.

However, according to the requirement of NMR sample preparation, typical1H NMR spectra require 5-25 mg of material, and 13C spectra require 50-100 mg of material. This amount of material will allow you to obtain a 1H spectrum in a few minutes or a 13C spectrum in 20-60 minutes. In this study, it is very difficult to obtain so high concentrations of patulin degradation products. Therefore, we did not do NMR to further verify the structures of the patulin degradation products.

In addition, LC-Q-TOF-MS/MS has been successfully used to analyze and identify metabolites and degradation products of food contaminants or organic substances by other groups (Bateman et al., 2007; Jiang et al., 2022; Lin et al., 2012; Mikolajczak et al., 2019; Mortishire-Smith et al., 2012; Pokar et al., 2020; Wang et al., 2011; Zhou et al., 2017). Accurate mass measurements from TOF generate the elemental composition of ions (molecules and fragments). Moreover, tandem mass spectrometry (MS/MS) provides complementary structural information through in-source fragmentation using ESI. Therefore, it was proven to be qualified for the identification of unknown compounds.

References

Bateman, K.P., Castro-Perez, J., Wrona, M., Shockor, J.P., Yu, K., Oballa, R., Nicoll-Griffith, D.A. MSE with mass defect filtering for in vitro and in vivo metabolite identification. Rapid Commun. Mass Spectrom. 2007, 21, 1485–1496.

Jiang, J., Hu, Z., Boucetta, H., Liu, J., Song, M., Hang, T., Lu, Y. Identification of degradation products in flumazenil using LC-Q-TOF/MS and NMR: Degradation pathway elucidation. J. Pharm. Biomed. Anal. 2022, 215, 114764.

Lin, Z.S., Le, J., Hong, Z.Y. Liquid chromatography-quadrupole-time-of-fight mass spectrometry and its application in the metabolism (metabolomics) research of traditional Chinese medicines. Chin. Pharm. J. 2012, 47, 401–405.

Mikolajczak, B., Fornal, E., Montowska. LC–Q–TOF–MS/MS Identification of specific non-meat proteins and peptides in beef burgers. Molecules 2019, 24(1), 18.

Mortishire-Smith, R.J., O’Connor, D., Castro-Perez, J.M., Kirby, J. Accelerated throughput metabolic route screening in early drug discovery using high-resolution liquid chromatography/quadrupole time-of-flight mass spectrometry and automated data analysis. Rapid Commun. Mass Spectrom. 2005, 19, 2659–2670.

Pokar, D., Sahu, A.K., Sengupta, P. LC-Q-TOF-MS driven identification of potential degradation impurities of venetoclax, mechanistic explanation on degradation pathway and establishment of a quantitative analytical assay method. J. Anal. Sci. Technol. 2020, 11, 54.

Wang, F., Xie, F., Xue, X.F., Wang, Z.D., Fan, B., Ha, Y.M. Structure elucidation and toxicity analyses of the radiolytic products of aflatoxin B1 in methanol-water solution. J. Hazard. Mater. 2011, 192, 1192–1202.

Zhou, N., Qian, Q., Qi, P., Zhao, J., Wang, C., Wang, Q. Identification of degradation products and process impurities from Tterbutaline sulfate by UHPLC-Q-TOF-MS/MS and in silico toxicity prediction. Chromatographia 2017, 80, 793–804.

(4) Comment: The authors have claimed that cysteine can be used to eliminate patulin while refused to test the notion in the real-life situation, showing subjective bias to what should be a more objective look at what is being done. Furthermore, the lack of the data in apple juice suggests such approach has not practical benefits.

Response: We are very sorry for failing to clarify the purpose of our study to the reviewer in the first response.

It is well known that patulin is an unsaturated γ-Lactones, which could react with cysteine based on the Michael addition reaction, a nucleophilic addition reaction, and formed patulin-cysteine adducts (Jones & Young, 1968; Fliege & Metzler, 2000). However, the Michael addition reaction (a nucleophilic addition reaction) is carried out spontaneously under near neutral or alkaline conditions (pH 6.5~7.4) (Fliege & Metzler, 2000; Lindroth & Wright, 1978). However, the pH values of apple juice range from 3.0 to 4.0 (it is usually around pH 3.5), it is difficult to form the patulin-cysteine adducts based on the Michael addition reaction between patulin and cysteine under high acidic condition (pH<4.6). Therefore, the low pH value of solution is the critical factor influencing the degradation of patulin by cysteine (Ma et al., 2021).

In order to improve the removal efficiency of patulin by cysteine in high acidic condition (pH 3.0~4.0), we found that heat-assisted treatment can significantly increase the degradation efficiency of patulin (Diao et al., 2021). However, the degradation products and mechanisms of patulin by heat-assisted cysteine in highly acidic condition (pH 3.5) are still unknown. Therefore, we used the pH 3.5 of malic acid solution (the final concentration of malic acid in simulated solution was about 0.10 mmol/L) to simulate the high acidic environment of apple juice, and explore the degradation products and mechanisms of patulin by heat-assisted cysteine.

In this study, our main objective is to isolate and identify the degradation products of patulin by the LC-Q-TOF-MS/MS analysis, and then to deduce the possible reaction mechanisms involved in the degradation of patulin by heat-assisted cysteine under the highly acidic condition (pH 3.5). In order to have a clearer, accurate and rapid understanding of the degradation mechanisms of patulin by heat-assisted cysteine under highly acidic condition, and reduce the interference of other substances (such as apple juice, which contains nutritional components, metal ions, vitamins, and polyphenols, and so on), we used the patulin (purity≥98.0%) and cysteine (purity≥99.0%) in simulated apple juice (pH 3.5) to react at the set conditions, and then to isolate and identify the degradation products of patulin by the LC-Q-TOF-MS/MS analysis.

References

Diao, E.J., Ma, K., Zhang, H., Xie, P., Qian, S.Q., Song, H.W., Mao, R.F., Zhang, L.M. Thermal stability and degradation kinetics of patulin in highly acidic conditions: impact of cysteine. Toxins 2021, 13, 662.

Fliege, R., Metzler, M. Electrophilic properties of patulin. N-acetylcysteine and glutathione adducts. Chem. Res. Toxicol. 2000, 13 (5), 373–381.

Jones, J.B., Young, J.M. Carcinogenicity of lactones. III. The reaction of unsaturated γ-lactones with L-cysteine. J. Med. Chem. 1968, 11 (6), 1176–1182.

Lindroth, S., Wright, A.V. Comparison of the toxicities of patulin and patulin adducts formed with cysteine. Appl. Environ. Microbiol. 1978, 35 (6), 1003–1007.

Ma, K., Diao, E., Zhang, H., Qian, S., Xie, P., Mao, R., Song, H., Zhang, L. Factors influencing the removal of patulin by cysteine. Toxicon 2021, 203, 51–57.

These are our responses to the questions and comments from the reviewer. Thank you very much for your suggestions on our manuscript. We appreciate for your warm work earnestly, and hope that our responses will meet with approval.
